# GeoNLF: Geometry guided Pose-Free
# Neural LiDAR Fields

**Weiyi Xue**[*]
Tongji University
xwy@tongji.edu.cn

**Zehan Zheng**[*]
Tongji University
zhengzehan@tongji.edu.cn

**Fan Lu**
Tongji University
lufan@tongji.edu.cn

**Haiyun Wei**
Tongji University
2311399@tongji.edu.cn

**Guang Chen**[†]
Tongji University
guangchen@tongji.edu.cn

**Changjun Jiang**
Tongji University
cjjiang@tongji.edu.cn

## Abstract

Although recent efforts have extended Neural Radiance Fields (NeRF) into LiDAR point cloud synthesis, the majority of existing works exhibit a strong dependence on precomputed poses. However, point cloud registration methods struggle to achieve precise global pose estimation, whereas previous pose-free NeRFs overlook geometric consistency in global reconstruction. In light of this, we explore the geometric insights of point clouds, which provide explicit registration priors for reconstruction. Based on this, we propose **Geo**metry guided **N**eural **L**iDAR **F**ields (GeoNLF), a hybrid framework performing alternately global neural reconstruction and pure geometric pose optimization. Furthermore, NeRFs tend to overfit individual frames and easily get stuck in local minima under sparse-view inputs. To tackle this issue, we develop a selective-reweighting strategy and introduce geometric constraints for robust optimization. Extensive experiments on NuScenes and KITTI-360 datasets demonstrate the superiority of **GeoNLF** in both novel view synthesis and multi-view registration of low-frequency large-scale point clouds.

## 1 Introduction

Neural Radiance Fields (NeRFs) [37] have achieved tremendous achievements in image novel view synthesis (NVS). Recent studies have extended it to LiDAR point cloud synthesis [23, 51, 67, 70], mitigating the domain gap to real data and far surpassing traditional methods. Nevertheless, the majority of existing works exhibit a strong dependence on known precise poses. In the domain of images, conventional approaches rely on Structure-from-Motion algorithms like COLMAP [48] to estimate poses, which are prone to failure with sparse or textureless views. As an alternative, recent works [6, 21, 31, 41] such as BARF [31] employ bundle-adjusting techniques to achieve high-quality NVS while simultaneously enhancing the precision of pose estimation.

However, the sparse nature of LiDAR point clouds and their inherent absence of texture information distinguish them significantly from images. Trivial bundle-adjusting techniques from the image domain become less applicable in this context, encountering the following challenges: (1) Outdoor LiDAR point clouds (*e.g.*, 2Hz, 32-beam LiDAR keyframes in Nuscenes [9]) exhibit temporal and spatial sparsity. NeRF easily overfits the input views without addressing the geometric inconsistencies caused by inaccurate poses. Consequently, it fails to propagate sufficient gradients for effective pose optimization. (2) Point clouds lack texture and color information but contain explicit geometric features. However, the photometric-based optimization scheme of NeRFs overlooks these abundant geometric cues within the point cloud, which hinders geometric-based registration.

---

[*] Equal contribution. [†] Corresponding author. Our code is availiable at https://github.com/ispc-lab/GeoNLF.

38th Conference on Neural Information Processing Systems (NeurIPS 2024).

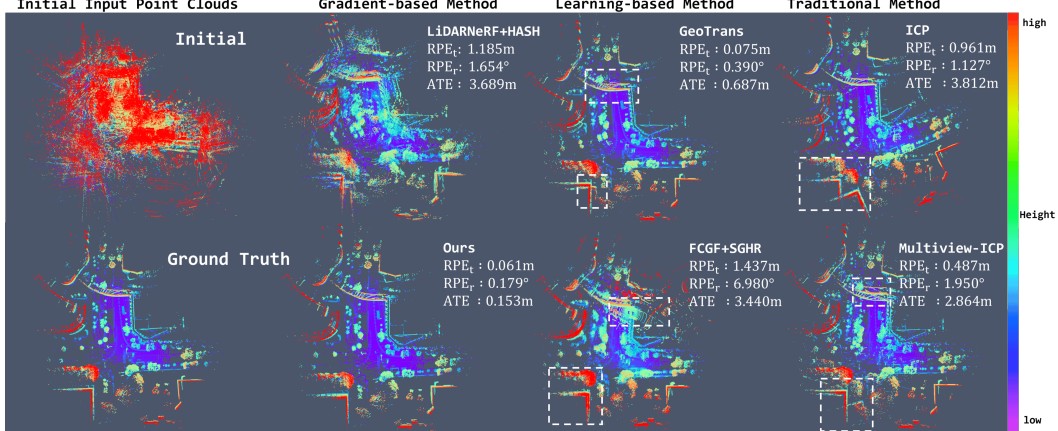

Figure 1: **Registration results.** Pairwise algorithms such as GeoTrans [44] and ICP [5] suffer from error accumulation and local mismatches. Multi-view methods like SGHR [54] and MICP [13] still manifest outlier poses. Previous gradient-based approaches LiDARNeRF-HASH [21] lack geometric consistency. Our method effectively avoids outlier frames and achieves superior registration accuracy.

An alternative to achieving pose-free LiDAR-NeRF is to employ point cloud registration (PCR) methods. Nonetheless, as the frequency of point cloud sequences decreases, the inter-frame motion escalates with a reduction in overlap. As presented in Fig. 1, pairwise and multi-view registration approaches may all trap in local optima and suffer from error accumulation, making it challenging to attain globally accurate poses. Hence, integrating local point cloud geometric features for registration with the global optimization of NeRF would be a better synergistic approach.

Furthermore, as demonstrated in [6, 53], the incorporation of geometric constraints significantly enhances the optimization of both pose and radiance fields. In the image domain, this process involves introducing additional correspondences or depth priors. However, most methods treat them solely as loss terms without fully exploiting them. In contrast, point clouds provide inter-frame correlations (*e.g.*, the closest point) for registration and explicit geometric information for reconstruction, presenting substantial advantages over images.

To this end, we propose **GeoNLF**, integrating LiDAR NVS with multi-view PCR for large-scale and low-frequency point clouds. Specifically, to address the suboptimality of global optimization and guide NeRF in the early pose optimization stage to avoid local minima, we regulate NeRF with a pure geometric optimizer. This module constructs a graph for multi-view point clouds and optimizes poses through graph-based loss. To reduce overfitting, we devised a selective-reweighting technique involving filtering out frames with outlier poses, thereby lessening their deleterious impacts throughout the optimization process. Additionally, to fully leverage the geometric attributes of point clouds, we introduced geometric constraints for point cloud modality rather than relying solely on the range map for supervision. Furthermore, our approach has demonstrated excellent performance in large-scale scenarios with sparse point cloud sequences at 2Hz, spanning hundreds of meters. To summarize, our main contributions are as follows:

(1) We propose **GeoNLF**, a novel framework for simultaneous large-scale multi-view PCR and LiDAR NVS. By exploiting geometric clues inside point clouds, **GeoNLF** couples geometric optimizer with neural reconstruction in the pose-free paradigm. (2) We introduce a selective-reweighting method to effectively alleviate overfitting, which presents excellent robustness across various scenarios. (3) Comprehensive experiments demonstrate **GeoNLF** outperforms state-of-the-art methods by a large margin on challenging large-scale and low-frequency point cloud sequences.

## 2   Background and Related Work

**Neural Radiance Fields.** NeRF [37] and related works have achieved remarkable achievements in NVS. Various neural representations [4, 10, 11, 22, 38], such as hash grids [38], triplanes [10, 22] and diverse techniques [39, 40, 55, 66] have been proposed to enhance NeRF's performance. Due to the lack of geometric information in images, some methods [16, 46, 59, 64] introduce depth prior or point clouds as auxiliary data to ensure multi-view geometric consistency. However, the geometric information and consistency encapsulated in point clouds are still not fully explored and utilized.

**Novel View Synthesis for LiDAR**. Traditional simulators [17, 27, 49] and explicit reconstruct-then-simulate [20, 28, 35] method exhibit large domain gap compared to real-world data. Very recently, a few studies have pioneered in NVS of LiDAR point clouds based on NeRF, surpassing traditional simulation methods. Among them, NeRF-LiDAR [68] and UniSim [62] require both RGB images as inputs. LiDAR-NeRF [51] and NFL [23] firstly proposed the differentiable LiDAR NVS framework, and LiDAR4D [70] further extended to dynamic scenes. However, most of these approaches still require a pre-computed pose of each point cloud frame and lack attention to geometric properties.

**Point Cloud Registration**. ICP [5] and its variants [45, 47, 43] are the most classic methods for registration, which rely on good initial conditions but are prone to falling into local optima. Learning-based method can be categorized into two schemes, i.e., end-to-end registration [65, 29, 24, 56, 1] and feature matching-based registration such as FCGF [14]. Recently, the specialized outdoor point cloud registration methods HRegNet [34] and HDMNet [61] have achieved excellent results. GeoTransformer [44] has achieved state-of-the-art in both indoor and outdoor point cloud registration. However, learning-based methods are data-driven and limited to specific datasets with ground truth poses, which requires costly pretraining and suffers from poor generalization.

Multiview methods are mostly designed for indoor scenes. Apart from Multiview-ICP [13, 7, 36], modern methods [2, 8, 52, 25] take global cycle consistency to optimize poses starting from an initial set of pairwise maps. Recent developments [19, 54, 3] such as SGHR [54] employ an iteratively reweighted least-squares (IRLS) scheme to adaptively downweight noisy pairwise estimates. However, their registration accuracy fundamentally depends on pairwise registration. The issues of pairwise methods for NVS still persist.

**Bundle-Adjusting NeRF**. iNeRF [63] and subsequent works [32, 15] demonstrated the ability of a trained NeRF to estimate novel view image poses through gradient descent. NeRFmm [58] and SCNeRF [50] extend the method to intrinsic parameter estimation. BARF [31] uses a coarse-to-fine reconstruction scheme in gradually learning positional encodings, demonstrating notable efficacy. Subsequent work HASH [21] adapts this approach on iNGP [38] through a weighted schedule of different resolution levels, further boosting performance. Besides, some studies have extended BARF to address more challenging scenarios, such as sparse input [53], dynamic scenes [33] and generalizable NeRF [12]. And [6, 53] uses monocular depth or correspondences priors for scene constraints, significantly enhancing the optimization of both pose and radiance fields. However, the aforementioned methods cannot be directly applied to point clouds or experience dramatic performance degradation when transferring. In contrast, our work is the first to introduce bundle-adjusting NeRF into LiDAR NVS task and achieve excellent results in challenging outdoor scenarios.

## 3 Methodology

We firstly introduce the pose-free Neural LiDAR Fields and the problem formulation of pose-free LiDAR-NVS. Following this, a detailed description of our proposed **GeoNLF** framework is provided.

**Pose-Free NeRF and Neural LiDAR Fields.** NeRF represents a 3D scene implicitly by encoding the density $\sigma$ and color $c$ of the scene using an implicit neural function $F_\Theta(x, d)$, where $x$ is the 3D coordinates and $d$ is the view direction. When synthesizing novel views, NeRF employs volume rendering techniques to accumulate densities and colors along sampled rays. While NeRF requires precise camera parameters, pose-free NeRF only uses images $\mathcal{I} = \{I_i | i = 0, 1..., N-1\}$ and treats camera parameters $\mathcal{E} = \{\mathcal{E}_s | s = 0, 1...N-1\}$ as learnable parameters similar to $\Theta$. Hence, the simultaneous update via gradient descent of $\mathcal{E}$ and $\Theta$ can be achieved by minimizing the error $\mathcal{L} = \sum_{i=0}^{N} \|\hat{I}_i - I_i\|_2^2$ between the rendered and ground truth image $\hat{I}, I$:

$$\Theta^*, \mathcal{E}^* = \arg\min_{\Theta, \mathcal{E}} \mathcal{L}(\hat{\mathcal{I}}, \hat{\mathcal{E}} \mid \mathcal{I}) \tag{1}$$

Following [70, 51], we project the LiDAR point clouds into range image, then cast a ray with a direction $d$ determined by the azimuth angle $\theta$ and elevation angle $\phi$ under the polar coordinate system: $d = (\cos\theta\cos\phi, \ \sin\theta\sin\phi, \ \cos\phi)^T$. Like pose-free NeRF, our pose-free Neural LiDAR Fields treats LiDAR poses as learnable parameters and applies neural function $F_\Theta$ to obtain a radiance depth $z$ and a volume density value $\sigma$. Subsequently, volume rendering techniques are employed to derive the pixel depth value $\hat{\mathcal{D}}$:

$$\hat{\mathcal{D}}(\mathbf{r}) = \sum_{i=1}^{N} T_i \left(1 - e^{-\sigma_i \delta_i}\right) z_i, \quad T_i = \exp(-\sum_{j=1}^{i-1} \sigma_j \delta_j) \tag{2}$$

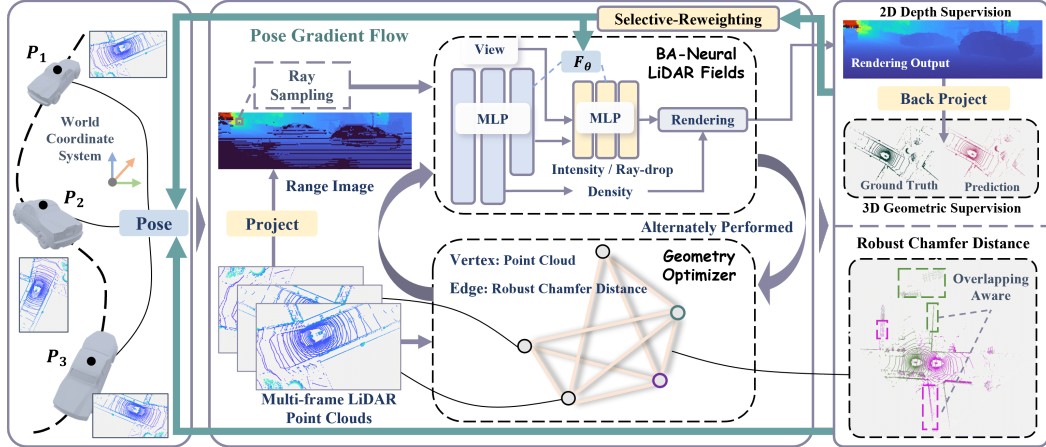

Figure 2: **Overview of our proposed GeoNLF.** We alternatively execute global optimization of bundle-adjusting neural LiDAR fields and graph-based pure geometric optimization. By integrating selective-reweighting strategy and explicit geometric constraints derived from point clouds, GeoNLF implements outlier-aware and geometry-aware mechanisms.

where $\delta$ refers to the distance between samples. We predict the intensity $\mathcal{S}$ and ray-drop probability $\mathcal{R}$ separately in the same way. Besides, our pose-free Neural LiDAR Fields adopted the Hybrid Planar-Grid representation from [70] for positional encoding $\gamma(x, y, z) = \mathbf{f}_{\text{planar}} \oplus \mathbf{f}_{\text{hash}}$.

$$\mathbf{f}_{\text{planar}} = \prod_{i=1}^{3} \text{Bilinear}(\mathcal{V}, \boldsymbol{x}), \mathcal{V} \in \mathbb{R}^{3 \times M \times M \times C}, \ \mathbf{f}_{\text{hash}} = \text{TriLinear}(\mathcal{H}, \boldsymbol{x}), \mathcal{G} \in \mathbb{R}^{M \times M \times M \times C} \quad (3)$$

where $\boldsymbol{x}$ is the 3D point, $\mathcal{V}, \mathcal{H}$ store the grid features with $M$ spatial resolution and $C$ feature channels. This encoding method benefits the representation of large-scale scenes[70].

**Problem Formulation**. In the context of large-scale outdoor driving scenarios, the collected LiDAR point cloud sequence $\mathcal{P} = \{\mathcal{P}_s | s = 0, 1, ..., N-1\}$ serves as inputs with a low sampling frequency. The goal of GeoNLF is to reconstruct this scene as a continuous implicit representation based on neural fields, jointly recovering the LiDAR poses $\mathcal{E} = \{\mathcal{E}_s | s = 0, 1, ..., N-1\}$ which can align all point clouds $\mathcal{P}$ globally.

## 3.1 Overview of GeoNLF Framework

In contrast to prior pose-free NeRF methods, our pipeline employs a hybrid approach to optimize poses. As shown in Fig. 2, the framework can be divided into two alternately executed parts: global optimization of bundle-adjusting neural LiDAR fields (Sec. 3.2) and graph-based pure geometric optimization (Sec. 3.3) with the proposed Geo-optimizer. In the first part, we adopt a coarse-to-fine training strategy [31] and extend it to the Hybrid Planar-Grid encoding [70]. In the second part, inspired by multi-view point cloud registration, we construct a graph between multiple frame point clouds and propose a graph-based loss. The graph enables us to achieve pure geometric optimization, which encompasses both inter-frame and global optimization. Furthermore, we integrate the selective-reweighting strategy (Sec. 3.4) into the global optimization. This encourages the gradient of outliers to propagate towards pose correction while lowering the magnitude transmitted to the radiance fields, thus mitigating the adverse effects of outliers during reconstruction. To ensure geometry-aware results, we additionally incorporate explicit geometric constraints derived from point clouds in Sec. 3.5.

## 3.2 Bundle-Adjusting Neural LiDAR Fields for Global Optimization

In the stage of global optimization, we optimize Neural LiDAR Fields while simultaneously back-propagating gradients to the pose of each frame. By optimizing our geometry-constrained loss, which will be detailed in Sec. 3.5, the pose is individually optimized to achieve global alignment.

**LiDAR Pose Representation**. In previous pose-free NeRF methods, poses are often modeled by $\boldsymbol{T} = [\boldsymbol{R} \mid \boldsymbol{t}] \in SE(3)$ with a rotation $\boldsymbol{R} \in SO(3)$ and a translation $\boldsymbol{t} \in \mathbb{R}^3$. Pose updates are computed in the special Euclidean Lie algebra $\mathfrak{se}(3) = \{\boldsymbol{\xi} = \begin{bmatrix} \boldsymbol{\rho} \\ \boldsymbol{\phi} \end{bmatrix}, \boldsymbol{\rho} \in \mathbb{R}^3, \boldsymbol{\phi} \in \mathfrak{so}(3)\}$ by

$\boldsymbol{\xi}' = \boldsymbol{\xi} + \Delta\boldsymbol{\xi}$, followed by the exponential map to obtain the transformation matrix $\boldsymbol{T}$:

$$\boldsymbol{T} = \exp(\boldsymbol{\xi}^\wedge) = \sum_{n=0}^{\infty} \frac{1}{n!}(\boldsymbol{\xi}^\wedge)^n = \begin{bmatrix} \sum_{n=0}^{\infty} \frac{1}{n!}(\boldsymbol{\phi}^\wedge)^n & \sum_{n=0}^{\infty} \frac{1}{(n+1)!}(\boldsymbol{\phi}^\wedge)^n \boldsymbol{\rho} \\ \mathbf{0}^T & 1 \end{bmatrix} \quad (4)$$

where $\xi^\wedge = \begin{bmatrix} \boldsymbol{\phi}^\wedge & \boldsymbol{\rho} \\ \mathbf{0}^T & 0 \end{bmatrix}$ and $\phi^\wedge$ is the antisymmetric matrix of $\phi$. Given a rotation vector $\boldsymbol{\phi} \in \mathfrak{so}(3)$, rotation matrix $\boldsymbol{R}$ can be obtained through the exponential map $\boldsymbol{R} = \exp(\boldsymbol{\phi}^\wedge) = \sum_{n=0}^{\infty} \frac{1}{n!}(\phi^\wedge)^n$. Simultaneously, we denote $\sum_{n=0}^{\infty} \frac{1}{(n+1)!}(\phi^\wedge)^n$ as $\boldsymbol{J}$. Then Eq. (4) can be rewritten as:

$$\boldsymbol{T} = \begin{bmatrix} \boldsymbol{R} & \textcolor{red}{\boldsymbol{J}}\boldsymbol{\rho} \\ \mathbf{0}^T & 1 \end{bmatrix} \quad (5)$$

Consequently, due to the coupling between $\boldsymbol{R} = \sum_{n=0}^{\infty} \frac{1}{n!}(\phi^\wedge)^n$ and $J = \sum_{n=0}^{\infty} \frac{1}{(n+1)!}(\phi^\wedge)^n$, the translation updates are influenced by rotation. Incorporating momentum may lead to non-intuitive optimization trajectories [32]. Therefore, we omit the coefficient $\boldsymbol{J}$ from the translation term. This approach enables updating the translation of the the center of mass and the rotation around the center of mass independently.

**Coarse-to-Fine Positional Encoding**. BARF[31]/HASH [21] propose to gradually activate high-frequency/high-resolution components within positional encoding. We further apply this approach to multi-scale planar and hash encoding [70] and found it also yields benefits in our large-scale scenarios. For the detailed formulation, we direct readers to reference [21].

### 3.3 Graph-based Pure Geometric Optimization

ICP [5] is a classic method for registration based on inter-frame geometric correlations. The essence of ICP lies in searching for the closest point as correspondence in another frame's point at each iteration, followed by using Singular Value Decomposition (SVD) to solve Eq. (6), then iteratively refining the solution. Nonetheless, ICP frequently converges to local optima (Fig. 1). In contrast, NeRF optimizes pose globally through the implicit radiance fields. However, it lacks geometric constraints and overlooks the strong geometric information inherent in the point cloud, leading to poor geometric consistency. As a consequence, both ICP and NeRF acting individually tend to converge to local optima. Our goal is to employ a hybrid method, utilizing NeRF for global pose optimization and integrating geometric information as an auxiliary support.

Drawing inspiration from ICP [5], we recognize that minimizing the Chamfer Distance (CD) is in line with the optimization objective of each step in ICP algorithm, as demonstrated in Eq. (7):

$$\mathbf{T}^* = \min_{\mathbf{T}} \sum_{\mathbf{p}_i \in \mathcal{P}} \min_{\mathbf{q}_i \in \mathcal{Q}} \|\mathbf{T} \cdot \mathbf{p}_i - \mathbf{q}_i\|_2^2 \quad (6)$$

$$\mathcal{L}_{(P,Q)} = \sum_{\mathbf{p}_i \in \mathbf{P}} w_i \min_{\mathbf{q}_i \in \mathbf{Q}} \|\mathbf{T}_\mathbf{P} p_i - \mathbf{T}_\mathbf{Q} q_i\|_2^2 + \sum_{\mathbf{q}_i \in \mathbf{Q}} w_i \min_{\mathbf{p}_i \in \mathbf{P}} \|\mathbf{T}_\mathbf{Q} q_i - \mathbf{T}_\mathbf{P} p_i\|_2^2 \quad (7)$$

where $q, p$ in point cloud $\mathbf{Q}, \mathbf{P}$ are homogeneous coordinates. $\mathbf{T}_P, \mathbf{T}_Q$ represent the transformation matrix to the world coordinate system. However, minimizing the original CD does not necessarily indicate improved accuracy due to the non-overlapping regions between point clouds. To alleviate this negative impact, we weight each correspondence based on Eq. (8), whereas $w_i$ in the original CD is normalized by the $\frac{1}{N}$, N is the number of points.

$$w_i = \frac{\exp(t/d_{clipped}^i)}{\sum_{i=1}^{N} \exp(t/d_{clipped}^i)}, \quad t = \text{scheduler}(t_0), \quad d_{clipped}^i = \max(\text{voxelsize}, d_i) \quad (8)$$

where $d_i$ denotes the distance between a pair of matching nearest neighbor points, $t$ is the temperature to sharpen the distribution of the $d_{clipped}^i$. The distance $d_i$ is clipped to the size of the downsampled voxel grid. This soft assignment can be considered as an approximately derivable version of weighted averaging. Eq. (7) will degenerate to the original CD when $t \to 0$, degrade to considering only correspondences with the minimum distance when $t \to \infty$. Considering the distance lacks practical significance in initial optimization, the scheduler is set as a linear or exponential function to vary $t$ from 0 to 0.5 as the optimization progresses. Building upon the above, as shown in Fig. 3, we

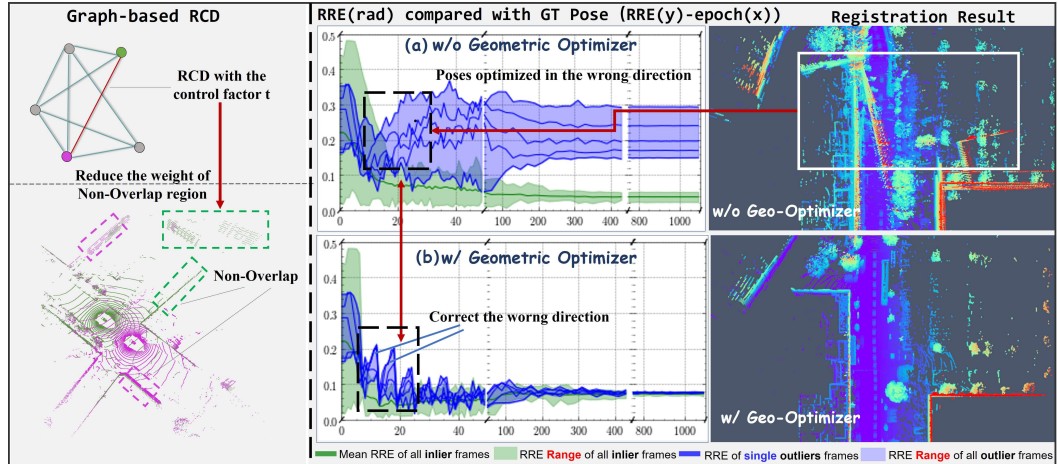

Figure 3: **Graph-based RCD (left)**. We introduce control factor $t$ in CD to diminish the weighting of non-overlapping regions between point clouds. **Geo-optimizer and its impact on pose optimization (right)**. Pose errors are reduced after each increase caused by NeRF's incorrect optimization direction. Comparison of (a) and (b) shows Geo-optimizer prevents incorrect pose optimization of NeRF.

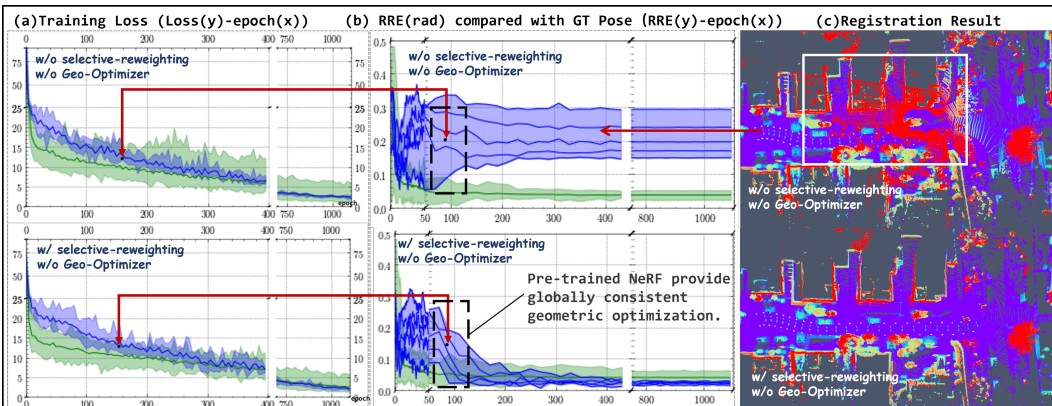

Figure 4: **Impact of selective-reweighting training strategy on pose optimization.** (a) Frames with outlier poses exhibit significantly higher losses. With selective-reweighting, outlier frames maintain a relatively higher loss without overfitting. (b) After several training iterations, the pre-trained outlier-aware NeRF can provide globally consistent geometric optimization for outlier frames.

approximate the registration objective by optimizing the Graph-based Robust Chamfer distance (G-RCD). Specifically, we construct a graph $(\mathcal{W}, \mathcal{Y})$, where each vertex $\mathcal{W}$ represents a set of points and each edge $\mathcal{Y}$ corresponds to proposed RCD via Eq. (7). We connect each frame with its temporally preceding $n$ frames to mitigate error accumulation in ICP [5]. Then RCD is calculated for all edges as Eq. (9), and $M$ denotes the number of frames in the sequence. Notably, in Eq. (7), G-RCD is computed using the global transform matrix, enabling direct gradient propagation of the Graph-based loss to the global transformation matrix of each frame.

$$\mathcal{L}_{graph} = \frac{1}{\left(nM - \frac{n(n+1)}{2}\right)} \sum_{(i,j) \in \mathcal{E}} \mathcal{L}_{(i,j)}, \tag{9}$$

**Discussion**. As illustrated in Fig. 3(b), insufficient geometric guidance leads to certain frame poses being optimized in the wrong direction. Geometric optimizer can address this issue by preventing pose updates strictly following NeRF and correcting wrong optimization directions that do not conform to global geometric consistency. This method involves externally modifying pose parameters and providing effective geometric guidance early in the ill-conditioned optimization process. Consequently, few iterations of graph-based RCD computation suffice to offer ample guidance for NeRF.

### 3.4 Selective-Reweighting Strategy for Outlier Filtering

In bundle-adjusting optimization, as shown in Fig. 4(a), we observed that frames with outlier poses present significantly higher rendering losses during the early stages of training. However, low

frequency and sparsity of point clouds result in quick overfitting of individual frames including outliers (*cf.* Fig. 4(a)(b)). This leads to minimal pose updates when the overall loss decreases, resulting in incorrect poses and inferior reconstruction. Inspired by the capabilities of NeRF in pose inference [63], we decrease the learning rate (`lr`) of neural fields for the top k frames with the highest rendering losses as Eq. (10), while keeping `lr` of poses unchanged. The strategy facilitates gradient propagation towards outlier poses, while the gradient flow to the radiance fields is concurrently diminished. Consequently, it's analogous to leveraging a pre-trained NeRF for outlier pose correction and lessens the adverse effects caused by outliers during the optimization process.

$$\text{lr}_{outliers} = (w_0 + l(1 - w_0))\text{lr}_{inliers} \quad (w_0 > 0) \tag{10}$$

Where $l \in [0, 1]$ denotes training progress. Akin to leaky ReLU [60], we set the reweighting factor $w_0$ to a relatively small value. $w_0$ increases as the process progresses, which ensures the network's ongoing learning from these frames and avoids stagnation.

### 3.5 Improving Geometry Constraints for NeRF

Point clouds encapsulate rich geometric features. However, solely supervising NeRF training via range images pixel-wise fails to fully exploit their potential, *e.g.*, normal information. Furthermore, the Chamfer distance can directly supervise the synthesized point clouds from a 3D perspective. Therefore, in addition to supervising via 2D range map, we propose directly constructing a three-dimensional geometric loss function between the generated point cloud and the ground truth point cloud. Unlike our Geo-optimizer, Eq. (11) imposes constraints between synthetic point clouds $\hat{P}$ and ground truth point clouds $P$:

$$\mathcal{L}_{CD} = \frac{1}{N_{\hat{P}}} \sum_{\hat{p}_i \in \hat{P}} \min_{p_i \in P} \|\hat{p}_i - p_i\|_2^2 + \frac{1}{N_P} \sum_{p_i \in P} \min_{\hat{p}_i \in \hat{P}} \|p_i - \hat{p}_i\|_2^2 \tag{11}$$

Based on the point correspondences established between $\hat{P}$ and $P$ as derived in Eq. (11), the constraint of normal can be formulated as minimizing:

$$\mathcal{L}_{normal} = \frac{1}{N_{\hat{P}}} \sum_{\hat{p}_i \in \hat{P}} \min_{p_i \in P} \|\mathcal{N}(\hat{p}_i) - \mathcal{N}(p_i)\|_1 + \frac{1}{N_P} \sum_{p_i \in P} \min_{\hat{p}_i \in \hat{P}} \|\mathcal{N}(p_i) - \mathcal{N}(\hat{p}_i)\|_1 \tag{12}$$

Thus, the normal loss is calculated between the synthetic point cloud and the ground truth point cloud to ensure more accurate normal vectors of the point cloud synthesized from NeRF. Moreover, we also employ 2D loss function to supervise NeRF as Eq. (13).

$$\mathcal{L}_r(\mathbf{r}) = \sum_{\mathbf{r} \in \mathbf{R}} \lambda_d \left\| \hat{D}(\mathbf{r}) - D(\mathbf{r}) \right\|_1 + \lambda_s \left\| \hat{\mathcal{S}}(\mathbf{r}) - \mathcal{S}(\mathbf{r}) \right\|_2^2 + \lambda_r \left\| \hat{\mathcal{R}}(\mathbf{r}) - \mathcal{R}(\mathbf{r}) \right\|_2^2 \tag{13}$$

where $D$ represents depth and $\mathcal{S}, \mathcal{R}$ represents intensity and ray-drop probabilities. Consequently, the loss for Neural LiDAR fields is weighted combination of the depth, intensity, ray-drop loss and 3D geometry constraints, which can be formalized as $\mathcal{L} = \mathcal{L}_r + \lambda_n \mathcal{L}_{normal} + \lambda_c \mathcal{L}_{CD}$.

## 4 Experiment

### 4.1 Experimental Setup

**Datasets and Experimental Settings.** We conducted experiments on two public autonomous driving datasets: NuScenes [9] and KITTI-360 [30] dataset, each with five representative LiDAR point cloud sequences. We selected 36 consecutive frames at 2Hz from keyframes as a single scene for NuScenes, holding out 4 samples at 9-frame intervals for NVS evaluation. KITTI-360 has an acquisition frequency of 10Hz. We used 24 consecutive frames sampled every 5th frame to match scene sizes of Nuscenes, holding out 3 samples at 8-frame intervals for evaluation. We perturbed LiDAR poses with additive noise corresponding to a standard deviation of $20 \deg$ in rotation and $3m$ in translation.

**Metrics.** We evaluate our method from two perspectives: pose estimation and novel view synthesis. For pose evaluation, we use standard odometry metrics, including Absolute Trajectory Error (ATE) and Relative Pose Error ($\text{RPE}_r$ in rotation and $\text{RPE}_t$ in translation). Following LiDAR4D [70] for NVS evaluation, we employ CD to assess the 3D geometric error and the F-score with 5cm error

| Method | Dataset | Point Cloud | | Depth | | | | | Intensity | | | | |
|---|---|---|---|---|---|---|---|---|---|---|---|---|---|
| | | CD↓ | F-score↑ | RMSE↓ | MedAE↓ | LPIPS↓ | SSIM↑ | PSNR↑ | RMSE↓ | MedAE↓ | LPIPS↓ | SSIM↑ | PSNR↑ |
| BARF-LN [31, 51] | Nuscenes | 1.2695 | 0.7589 | 8.2414 | 0.1123 | 0.1432 | 0.6856 | 20.89 | 0.392 | 0.0144 | 0.1023 | 0.6119 | 26.2330 |
| HASH-LN [21, 51] | | 0.9691 | 0.8011 | 7.8353 | 0.0441 | 0.1190 | 0.6543 | 20.6244 | 0.0459 | 0.0135 | 0.0954 | 0.6279 | 26.8870 |
| GeoTrans [44, 51] | | 4.1587 | 0.2993 | 10.7899 | 2.1529 | 0.1445 | 0.3671 | 17.5885 | 0.0679 | 0.0256 | 0.1149 | 0.3476 | 23.6211 |
| **GeoNLF (Ours)** | | **0.2408** | **0.8647** | **5.8208** | **0.0281** | **0.0727** | **0.7746** | **22.9472** | **0.0378** | **0.0100** | **0.0774** | **0.7368** | **28.6078** |
| BAR-LN [31, 51] | KITTI-360 | 3.1001 | 0.6156 | 7.5767 | 2.0583 | 0.5779 | 0.2834 | 22.5759 | 0.2121 | 0.1575 | 0.7121 | 0.1468 | 11.9778 |
| HASH-LN [21, 51] | | 2.6913 | 0.6082 | 6.3005 | 2.1686 | 0.5176 | 0.3752 | 22.6196 | 0.2404 | 0.1502 | 0.6508 | 0.1602 | 12.9286 |
| GeoTrans [44, 51] | | 0.5753 | 0.8116 | 4.4291 | 0.2023 | **0.3896** | 0.5330 | **25.6137** | 0.2709 | 0.1589 | 0.5578 | 0.2578 | 12.9707 |
| **GeoNLF (Ours)** | | **0.2363** | **0.9178** | **4.0293** | **0.1009** | 0.3900 | **0.6272** | 25.2758 | **0.1495** | 0.1525 | **0.5379** | **0.3165** | **16.5813** |

Table 1: **NVS Quantitative Comparison on Nuscenes and KITTI-360.** We compare our method to different types of approaches and color the top results as best and second best . All results are averaged over the 5 sequences.

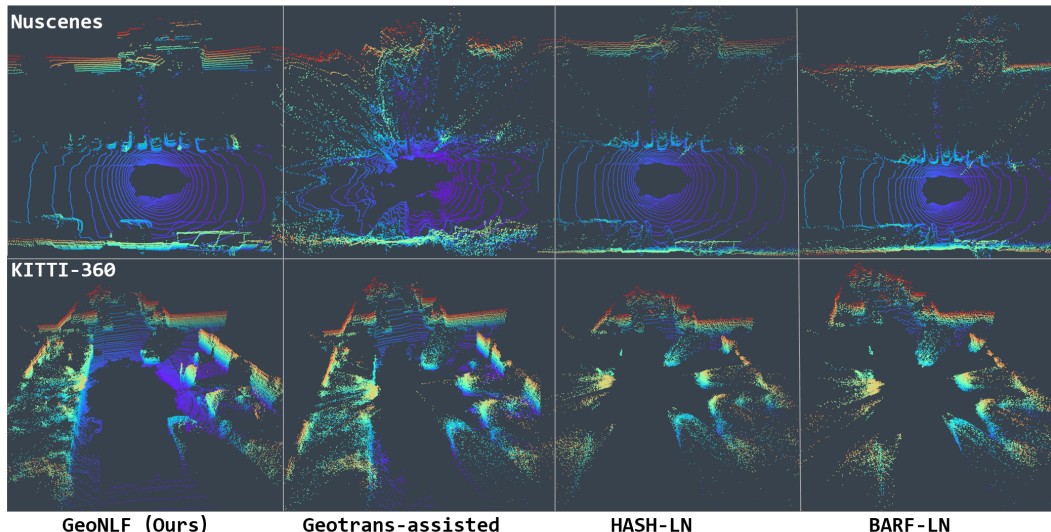

Figure 5: **Qualitative comparison of NVS.** We compared GeoNLF with other pose-free methods and GeoTrans-assisted NeRF. Especially, GeoTrans fails on Nuscenes due to the inaccurate poses.

threshold. Additionally, we use RMSE and MedAE to compute depth and intensity errors in projected range images, along with LPIPS [69], SSIM [57], and PSNR to measure overall variance.

**Implementation Details.** The entire point cloud scene is scaled within the unit cube space. The optimization of GeoNLF is implemented on Pytorch [42] with Adam [26] optimizer. All the sequences are trained for 60K iterations. Our Geometry optimizer's `lr` for translation and rotation is the same as the `lr` for pose in NeRF with synchronized decay. We use the coarse-to-fine strategy[31, 21], which starts from training progress 0.1 to 0.8. The reweight coefficient for the top-5 frames linearly increases from 0.15 to 1 during training. After every $m_1$ epoch of bundle adjusting global optimization, we proceed with $m_2$ epoch of pure geometric optimization, where $m_2/m_1$ decrease from 10 to 1.

## 4.2 Comparison in LiDAR NVS

We compare our model with BARF [31] and HASH [21], both of which use LiDAR-NeRF[51] as backbone. For PCR-assisted NeRF, we opt to initially estimate pose utilizing pose derived from GeoTrans [44], which is the most robust and accurate algorithm among other PCR methods in our experiments. And subsequently we leverage LiDAR-NeRF [51] for reconstruction. For all Pose-free methods, we follow NeRFmm[58] to obtain the pose of test views for rendering. The quantitative and qualitative results are in Tab. 1 and Fig. 5. Our method achieves high-precision registration and high-quality reconstruction across all sequences. However, baseline methods fail completely on certain sequences due to their lack of robustness. Please refer to Fig. 7 for details. Ultimately, our method excels in the reconstruction of depth and intensity, as evidenced by 7.9% increase in F-score on Nuscenes and 13.1% on KITTI-360 compared to the second best result.

| Method | NuScenes | | | KITTI-360 | | |
|---|---|---|---|---|---|---|
| | $RPE_t$(cm)↓ | $RPE_r$(deg)↓ | ATE(m)↓ | $RPE_t$(cm)↓ | $RPE_r$(deg)↓ | ATE(m)↓ |
| ICP [5] | 15.410 | 0.647 | 1.131 | 30.383 | 1.019 | 1.894 |
| MICP [51] | 38.84 | 1.101 | 2.519 | 35.584 | 1.419 | 1.483 |
| HRegNet [34] | 120.913 | 2.173 | 7.815 | 290.16 | 9.083 | 7.423 |
| SGHR [54] | 100.98 | 0.699 | 9.557 | 95.576 | 0.906 | 2.539 |
| GeoTrans [44] | 16.097 | 0.363 | 0.892 | 6.081 | 0.213 | 0.246 |
| BARF-LN [51, 31] | 210.331 | 0.819 | 5.244 | 199.74 | 2.203 | 2.763 |
| HASH-LN [51, 21] | 180.282 | 0.832 | 4.151 | 196.791 | 2.171 | 2.666 |
| **GeoNLF (Ours)** | **7.058** | **0.103** | **0.228** | **5.449** | **0.205** | **0.170** |

Table 2: **Pose estimation accuracy comparison.**

| Method | Point Cloud | Depth | Intensity | Pose | | |
|---|---|---|---|---|---|---|
| | CD↓ | PSNR ↑ | PSNR ↑ | $RPE_t$(cm)↓ | $RPE_r$(deg)↓ | ATE(m)↓ |
| w/o G-optim | 0.6180 | 21.3211 | 25.8551 | 54.72 | 0.283 | 1.328 |
| w/o RCD | 0.2711 | 21.1323 | 26.7232 | 8.476 | 0.163 | 0.332 |
| w/o SR | 0.2654 | 21.1096 | 26.5269 | 8.124 | 0.156 | 0.264 |
| w/o $L_{3d}$ | 0.2877 | 21.7128 | 28.5210 | 7.273 | 0.124 | 0.234 |
| **GeoNLF** | **0.2363** | **22.9472** | **28.6078** | **7.058** | **0.103** | **0.228** |

Table 3: **Ablation study on Nuscenes.**

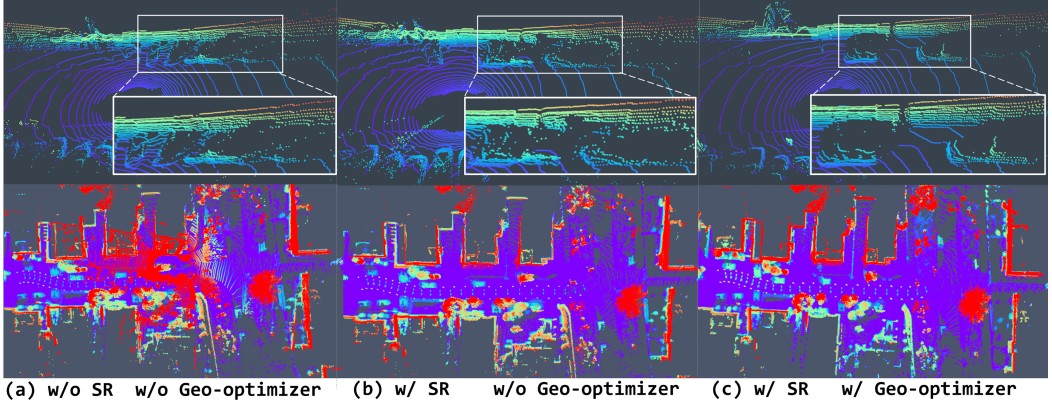

(a) w/o SR   w/o Geo-optimizer        (b) w/ SR   w/o Geo-optimizer        (c) w/ SR   w/ Geo-optimizer

Figure 6: **Qualitative results of ablation study.** We present the NVS and Registration results in the first and second rows. Outlier frames emerged w/o SR or w/o G-optim.

## 4.3 Comparison in Pose Estimation

We conduct comprehensive comparisons of GeoNLF with pairwise baselines, including traditional method ICP [5], learning-based GeoTrans [44] and outdoor-specific HRegNet [34], as well as multi-view baselines MICP [13] and learning-based SGHR [54]. For pairwise methods, we perform registration between adjacent frames in an Odometry-like way. For SGHR, we utilize FCGF [14] descriptors followed by RANSAC [18] for pairwise registration. The estimated trajectory is aligned with the ground truth using $\mathrm{Sim}(3)$ with known scale.

**GeoNLF** outperforms both the registration and pose-free NeRF baselines. Quantitative and Qualitative results are illustrated in Tab. 2 and Fig. 1. As depicted in Fig. 1, most registration methods fail to achieve globally accurate poses and **completely fail in some scenarios**, leading to massive errors in average results. Significant generalization issues arise for learning-based registration methods due to potential disparities between testing scenarios and training data, including differences in initial pose distributions. This challenge is particularly pronounced in HRegNet [34]. While the transformer model GeoTrans [44] with its higher capacity offers some alleviation to the issue, it remains not fully resolved.

## 4.4 Ablation Study

In this Section, we analyze the effectiveness of each component of GeoNLF. The results of ablation studies are shown in Tab. 3. **(1) Geo-optimizer.** When training GeoNLF w/o geo-optimizer (w/o G-optim), pose optimization may initially converge towards incorrect directions. Excluding geo-optimizer in GeoNLF results in decreased pose accuracy and reconstruction quality. **(2) Control factor of graph-based RCD.** Although geo-optimizer is crucial in the early stages of optimization, we find that using the original CD limits the accuracy of pose estimation. Removing the control factor (w/o RCD) leads to decreased pose estimation accuracy due to the presence of non-overlapping regions. **(3) Selective-reweighting (SR) strategy.** As presented in Figs. 4 and 6 and Tab. 3, outlier frames cause GeoNLF w/o SR strategy to overlook multi-view consistency, adversely affecting reconstruction quality. **(4) Geometric constraints.** Removing the 3D constraints (w/o $L_{3d}$) results in a decline in CD due to the photometric loss's inability to adequately capture geometric information.

## 4.5 Limination

Despite the fact that GeoNLF has exhibited exceptional performance in PCR and LiDAR-NVS on challenging scenes, it is not designed for dynamic scenes, which is non-negligible in autonomous

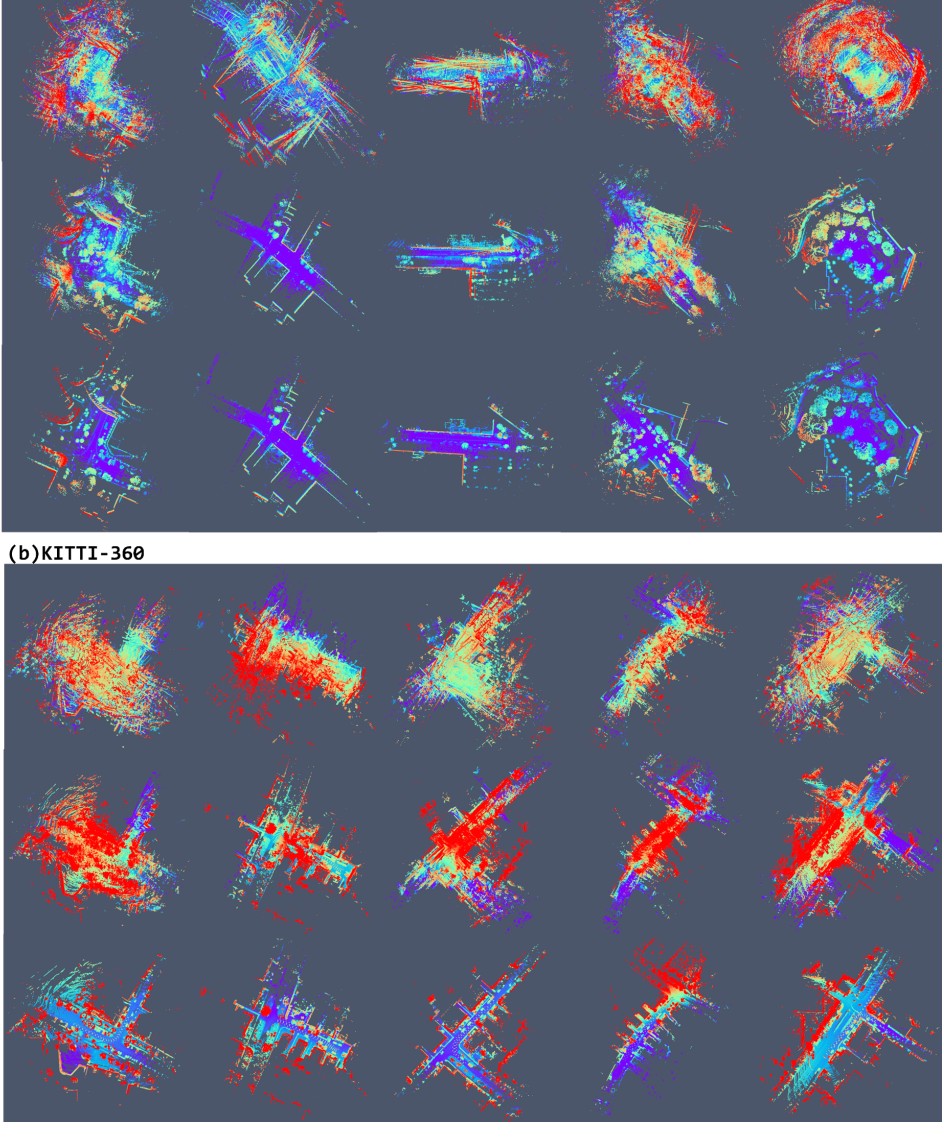

**(a)Nuscenes**

**(b)KITTI-360**

Figure 7: **Qualitative registration results of HASH-LN and GeoNLF on Nuscenes and KITTI-360 dataset.** The first row contains original inputs, the second row shows the results of HASH-LN, and the third row displays the results of GeoNLF.

driving scenarios. Additionally, GeoNLF targets point clouds within a sequence, relying on the temporal prior of the point clouds.

## 5    Conclusion

We introduce GeoNLF for multi-view registration and novel view synthesis from a sequence of sparsely sampled point clouds. We demonstrate the challenges encountered by previous pairwise and multi-view registration methods, as well as the difficulties faced by previous pose-free methods. Through the utilization of our Geo-Optimizer, Graph-based Robust CD, selective-reweighting strategy and geometric constraints from 3D perspective, our outlier-aware and geometry-aware GeoNLF demonstrate the promising performance in both multi-view registration and NVS tasks.

## 6    Acknowledgments

This work was supported by the National Key Research and Development Program of China (No.2024YFE0211000), in part by the National Natural Science Foundation of China (No. 62372329), in part by Shanghai Scientific Innovation Foundation (No.23DZ1203400), in part by Tongji-Qomolo Autonomous Driving Commercial Vehicle Joint Lab Project, and in part by Xiaomi Young Talents Program.

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
