# OpenReview forum: "GeoNLF: Geometry guided Pose-Free Neural LiDAR Fields"
_NeurIPS.cc/2024/Conference — NeurIPS 2024 poster_

### Official Review · Reviewer_TE2m · 2024-06-12

**Soundness:** 3
**Presentation:** 3
**Contribution:** 2
**Rating:** 6
**Confidence:** 3

**Summary:**

This paper presents GeoNLF, a new approach to pose-free large-scale point-cloud registration (PCR) and LiDAR novel view synthesis (NVS). To address the limitation of both ICP-based and NeRF-based methods, the authors proposed a hybrid approach involving both robust ICP and global optimisation of NeRF. Experiments show the proposed method achieved better pose estimation and NVS results compared to previous SOTA methods.

**Strengths:**

1. The paper is well-written and easy to follow.
2. The combination of geometric-based ICP and direct NeRF optimisation is novel and the explanation is very clear.
3. Fig. 3 and Fig. 4 are very informative in showing how the proposed geometric optimisation and selective re-weighting helps the optimisation process.
4. The experimental results are promising.

**Weaknesses:**

1. The proposed approach consists of so many building blocks which makes it a complicated system to work. Besides, from Tab. 3, it seems that except the geometric optimizer which contributes a lot to the final result, all other components seem to only produce very small improvement.

2. How long does the optimisation take? It would be good to show some run-time comparison and analysis.

**Questions:**

Please see the Weakness.

**Limitations:**

Yes.

---

> ### Author Rebuttal · Authors · 2024-08-07
>
> We greatly appreciate your positive feedback on our paper's presentation and the novelty of our method. Below are the responses to your concerns.
>
> **[Q1]: The issue of the algorithm's complexity and the role of the components(SR and geometric constraints).**
>
> **1. Complexity of Our Algorithm.** Our method combines LiDAR-NeRF with a Geo-optimizer, which constitutes the two main modules of our model. The Selective-Reweighting (SR) strategy (Sec 3.4) and the 3D geometric constraints (Sec 3.5) do not actually complicate the model structure and almost do not increase the optimization time (please kindly refer to global rebuttal for more details).
> Specifically, the SR strategy adjusts the learning rate for different frames during optimization to avoid overfitting frames with high loss during training. The geometric constraints introduce a geometry-based loss function to ensure that the synthetic point cloud possesses more accurate geometric features.
>
> **2. The Role of the Components.** We appreciate your inquiry regarding the roles of these two strategies(SR and geometric constraints). Given that the Geo-optimizer is crucial for the success of registration, it significantly enhances performance. Without this core module, the remaining two optimization strategies struggle to exert their influence effectively. However, these strategies play pivotal roles in specific aspects and metrics. We will now provide detailed explanations:
>
> **2.1. SR Strategy.** SR strategy aims to prevent overfitting of NeRF to individual frames and plays a crucial role in specific sequences. It primarily addresses extreme cases where most frame poses are optimized well, except for a few overfitted outliers. However, in most sequences, the significant improvement in registration success with NeRF+Geo-optimizer reduces extreme cases. Another factor to consider is that our ablation experiments are averaged over five Nuscenes sequences, which may average out its effects on certain frames. Considering its effectiveness becomes particularly evident in specific sequences, especially when outlier frames appear during the optimization process, we conducted an ablation study on the specific sequence (KITTI-360, Seq-3) to demonstrate its effectiveness, as shown in Fig. (2)(a) in the attached PDF and table below.
>
> | | CD$\downarrow$     | $\mathrm{PSNR_{depth}}\uparrow$ | $\mathrm{PSNR_{intensity}}\uparrow$ | $\mathrm{RPE_t}\downarrow$(cm) | $\mathrm{RPE_r}\downarrow$(deg) | $\mathrm{ATE}\downarrow$(m) |
> |-|-|-|-|-|-|-|
> | w/ SR  | 0.0510 | 28.5674|18.0307| **7.333** | **0.118** | **0.277** |
> | w/o SR | 0.0607 | 27.7158|17.2744| **69.570** | **0.667** | **1.421** |
>
> As shown in the table, in the absence of the SR strategy, the pose metrics exhibit a sharp decline due to the presence of outlier frames. Additionally, the negative impact of the outlier frame on NeRF optimization significantly contributes to a comprehensive degradation in NVS performance.
>
> **In summary, the SR strategy can enhance the overall performance of the model by preventing NeRF from overfitting high-loss frames in the early stages and enables our algorithm to be more robust across various sequences. Importantly, this strategy introduces no additional computational burden in optimization time and does not increase system complexity, thereby demonstrating practical utility. Furthermore, we believe that this simple yet effective strategy will also be beneficial for other pose-free NeRF works.**
>
> **2.2. 3D Geometric Constraints.** Regarding 3D geometric constraints, we acknowledge their marginal improvement in pose optimization accuracy and the 2D evaluation metrics of NVS. Their impact primarily manifests in optimizing the 3D geometric features of the synthesized point cloud, offering specific practical significance. Primarily, our geometric constraints supervise the 3D aspects of the synthesized point cloud through a dedicated loss function, aiming for enhanced geometric fidelity. Therefore, the improvements from geometric constraints are primarily reflected in 3D point cloud error metrics, **notably reducing CD (a 17.8\% reduction).** This indicates that geometric constraints help align the distribution of our model-synthesized point clouds closer to the ground truth point clouds in 3D space.
>
> Additionally, since our geometric constraints primarily affect the 3D domain, their impact on the evaluation metrics focusing on 2D range maps in NVS is not significantly pronounced. Hence, we illustrate the effectiveness of our geometric constraints from a 3D perspective.
>
> As shown in Fig. (2)(b) in the attached PDF, since previous LiDARNeRF[52] is optimized based on 2D range images, it effectively performs optimization from the front view perspective. Using 2D-based loss without our geometric constraints can still yield satisfactory results from the front view (compare Fig. (2) b.1 and b.2). **However, in the bird's-eye view, the reconstruction quality significantly deteriorates without our geometric constraints(compare Fig. (2) b.3 and b.4). This is because our geometric constraints provide supervision from a 3D perspective, addressing the limitations of 2D perspective supervision. As a result, our synthesized point cloud exhibits more realistic geometric features, such as smoother and more realistic surfaces.**
>
> **[Q2]: Run-time Comparison and Analysis.** Please refer to global rebuttal.

---

### Official Review · Reviewer_fgnm · 2024-07-01

**Soundness:** 4
**Presentation:** 4
**Contribution:** 3
**Rating:** 7
**Confidence:** 4

**Summary:**

In this submission the authors propose a novel strategy to jointly optimize a Neural LIDAR Field and the corresponding LIDAR poses. They achieve this by utilizing a hybrid strategy of alternating pure graph-based geometric optimization and global optimization of the Neural Field. Additionally they introduce a new selective reweighting strategy that steers gradients in the Neural Field towards outlier poses and new geometric losses that improve geometric reconstruction. They perform experiments on two datasets and achieve better results than relevant baselines in both reconstruction and registration.

**Strengths:**

1. The proposed method with its hybrid optimisation and other components is novel and very interesting and achieves impressive results relevant to the field of LIDAR simulation.
2. The paper is well written and gives deep insights into the optimisation procedure with well designed figures (e.g. Figure 3&4). The concepts are very clearly laid out and supported by experimental evidence and ablations. Some sentences could be improved but this is very minor and does not detract from the overall paper.
3. It is very hard to find technical faults in this paper. The experiments are comprehensive and consider relevant baselines and the introduced components are backed up with mathematical foundations quite well.

**Weaknesses:**

In order of severity:

1. Is it truly Pose-Free? The starting point for the LIDAR pose optimization is not completely random but they are perturbed from Ground truth by noise with a standard deviation of 20deg in rotation and 3m in translation (line 235). This means that some form of reasonable initialization is necessary for the method, which would have to be obtained from somewhere. How to obtain this initialization is not discussed in the paper.
2. The motivation for why this approach is necessary is unclear. The setting of obtaining sparse LIDAR scans without at least poses derived from GPS or vehicle odometry seems unrealistic, as well as the assumptions that cameras would not be available given that they are significantly cheaper than LIDAR Scanners. This subtracts from the otherwise strong paper because the setting and significance of the results cannot be properly contextualized. It is also not clear from the Introduction what motivation the simulation of LIDAR data in a Neural Field has and what potential applications are enabled by the proposed method.
3. While the authors report standard pose registration metrics (RPE & ATE) they do not consider Inlier Ratio, Feature Matching Recall or Registration Recall as metrics. This makes a comparison to prior work such as GeoTransformer [45] more complicated since these are the main metrics which they evaluate on in their paper. ([45] Qin, Z., Yu, H., Wang, C., Guo, Y., Peng, Y., & Xu, K. (2022). Geometric transformer for fast and robust point cloud registration. In Proceedings of the IEEE/CVF conference on computer vision and pattern recognition (pp. 11143-11152))
4. Outlier/Inlier poses and frames are discussed in Figure 3 and for example line 208 but it is never clarified how their inlier/outlier status is determined. This makes reproducibility harder and should be clarified either in the paper or supplementary.
5. In section 3.5 the authors introduce a new loss between generated and ground truth point clouds. It is not mentioned how the generated point cloud is obtained from the Neural Field, which is important to know since this determines how the gradient from the loss is backpropagated through this mechanism. This should also be clarified in the final version or supplementary.
6. In Equation 13 the variable P is very confusing since in the equations above it represents a point cloud and then here probably refers to the ray-drop loss. This is in contrast to line 119 & 120 where intensity and ray-drop probability are introduced as $S$ and $R$ respectively. This should be made coherent with each other or at least clarified.
7. In line 113-115 the authors explain how the direction of rays is determined but do not explain how they represent translation in their polar coordinate system. This could also be clarified in the supplementary.


Minor comments:
 - Line 16 NeRFs has achieved → NeRFs have achieved
 - Line 49 In furtherance of mitigating overfitting → To reduce overfitting. Clarity > Complicated Words.
 - Line 301 demonstrate the promising performance → demonstrate promising performance.

**Questions:**

To focus discussion about the weaknesses raised above here are some detailed questions:
1. How can the initial (bad) registration of the point clouds be obtained or is there a way this step can be removed ? Can the point cloud poses be initialized completely randomly or all as identity?
2. Is there some reason (e.g. Privacy concerns) that prohibit the joint capture of Image data via cameras and LIDAR scans? This would enable standard Structure from Motion approaches for pose estimation.
3. What is the overall motivation of representing LIDAR in a Neural Field and what benefits are gained by this? Are there downstream applications enabled by the proposed method?
4. Can you report the Inlier Ratio, Feature Matching Recall or Registration Recall ? If not, why can this not be reported on this data?

**Limitations:**

The authors have addressed the limitations of their work with regards to dynamic scenes, since they only consider static scenes.
Societal impact is not discussed.

---

> ### Author Rebuttal · Authors · 2024-08-07
>
> We greatly appreciate your careful reading and your questions provide valuable insights. We are pleased to address your concerns and engage in further discussion with you.
>
> **[Q1]: Initialization of Poses.** We follow previous works BARF[32] and NeRFmm[59], which are classical in the pose-free NeRF domain, to set up experiments. This setting is practical in autonomous driving scenarios, where pose estimations are not always accurate. Existing datasets with pose sensors(GPS, IMU), require calibration via time-consuming registration methods. Without initial pose information or with sparse inputs, SLAM or registration errors are more significant. Thus, adding random perturbations to each frame is a reasonable test of algorithm performance.
>
> Nevertheless, we also perform experiments on Nuscenes with all initial poses set to identity, and our algorithm remains effective, as shown in Fig. (1) and Tab. (1) of the attached PDF. Additionally, you may run the demo we provide by adding the flag '--no\_gt\_pose', which implies that all poses are initialized to identity. For practical applications, initialization can use either the identity matrix, coarse point cloud registration results, or GPS/IMU information, especially in outdoor scenes.
>
> **[Q2]: Reasons for Prohibiting the Joint Capture of Image and LiDAR Scans.** Firstly, as noted in L21-23, pose-free NeRF in camera domain approaches aim to eliminate NeRF's reliance on SfM algorithms, which are time-consuming and often fail in homogeneous regions and fast-changing view-dependent appearances. Thus, the primary motivation for pose-free NeRFs is to enable reconstruction without relying on time-consuming SfM computations and avoiding the instability of SfM failures. Additionally, directly using point cloud registration algorithms might be a better choice. This is because 3D point cloud registration, in most cases, achieves more accurate poses compared to 2D-based registration. However, as shown in Fig.(1) of the main text and L33-38, pairwise and multiview point cloud registration may still be prone to local optima and error accumulation, making it difficult to achieve globally accurate poses.
>
> In summary, poses obtained from GPS/IMU are often not sufficiently accurate(but can serve as a "bad initialization"), both image-based SfM and point cloud registration algorithms suffer from error accumulation and instability due to registration failures, and introduce extra computational overhead in reconstruction. Therefore, to achieve high-quality and efficient reconstruction, pose-free NeRFs are of practical significance.
>
> Besides, We commit to a LiDAR-only framework, aiming at achieving efficient reconstruction and NVS of point clouds without relying on images and accurate poses. Therefore, it has stronger applicability in LiDAR NVS.
>
> **[Q3]: Motivation and Benefit of Using NeRF and Downstream Application.** Traditional simulators and explicit reconstruct-then-simulate methods exhibit a large domain gap compared to real-world data. Moreover, traditional simulators like CARLA simulate
> within virtual environments, exhibiting diversity limitations and a reliance on costly 3D assets. Explicit reconstruction methods like LiDARsim[36] need to reconstruct mesh surfel representations, which are challenging for recovering precise surfaces in large-scale complex scenes. Additionally, NeRF inherently applies to the modeling of intensity and ray drop of LiDAR and does not require an intermediate modality for reconstruction, thereby eliminating the domain gap. Thus, some methods such as NFL[24] and LiDAR-NeRF[52], utilize NeRF to represent LiDAR data but need accurate poses.
>
> Besides, we appreciate your attention to the applications of our method.
> (1) Autonomous Driving Simulation: As shown in Fig. (3) of the attached PDF, we can conduct autonomous driving simulations in real-world scenarios rather than in simulators like CARLA, which is a prominent research field in autonomous driving. (2) Simulating point clouds from different LiDAR sensors: Our method allows for the adjustment of LiDAR sensors during NVS process. This flexibility facilitates simulations with various LiDAR configurations.
>
> Additionally, unlike previous methods that require accurate poses from multiple sensors, our approach circumvents this complex and time-intensive process, offering substantial practical benefits. Further applications can be referenced in LiDAR4D[71] and we look forward to discussing them with you in the future discussion.
>
> **[Q4]: Pose Registration metrics.**
> Registration Recall(RR) can be computed on this data. However, inlier Ratio and Feature Matching Recall cannot be reported on this data because these metrics are only applicable to feature-based point cloud registration methods, and our method does not require the computation of point features. Moreover, ATE and RPE are widely adopted in many pose-free NeRF works, such as Nope-NeRF[6] for evaluating the registration accuracy of a sequence of image/point clouds. However, RR is typically applied in pairwise registration, and is challenging to measure the registration performance of a sequence, because error accumulation in a sequence can prevent it from accurately reflecting registration performance. We realized that it is meaningful to compute the RR of adjacent frames by comparing  **gt relative poses** with predicted **relative poses**. The results are shown in Tab. (2) of the attached PDF.
>
> **[Weekness1(W1)]/ W[2]/ W[3].** Please refer to Q1/ Q2,Q3/ Q4.
>
> **[W4.]: Determining Outliers.** We directly select the top k frames with the highest training loss in each epoch as outliers without any priors (L206). The reason and effectiveness can be referenced in L201.
>
> **[W5, W6].** Please refer to the global rebuttal.
>
> **[W7]: Translation Representation.** We project the point cloud into a range image and each pixel on the image is represented as a ray, where the translation is the ray origin in the world coordinate system.

---

> > ### Comment · Reviewer_fgnm · 2024-08-09
> > **Comments on Rebuttal**
> >
> > Dear Authors,
> >
> > thank you for the comprehensive rebuttal.
> > I read it and my questions were addressed well.
> >
> > I would strongly encourage the authors to include the additional explanations provided here for Q2, Q3 in the introduction section of the final version of the paper, since this motivation was (in my opinion) missing in the initial submission.
> >
> > The additional experiments and clarifications provided in the rebuttal would also be nice to see in the final paper.
> >
> > I will raise my score to Accept.

---

> > > ### Author Response · Authors · 2024-08-10
> > > **Thanks for your comments!**
> > >
> > > We greatly appreciate your suggestions, which have enriched the completeness of our work and provided valuable insights. In the final paper, we will revise the introduction based on your recommendations, particularly regarding the selection of sensors and the background on LiDARNeRF, and we will include the experiments in the supplementary material. Thank you again for your recognition.

---

### Official Review · Reviewer_PTin · 2024-07-09

**Soundness:** 3
**Presentation:** 1
**Contribution:** 3
**Rating:** 5
**Confidence:** 4

**Summary:**

The paper proposes GeoNLF, a method for pose-free Lidar point cloud novel view synthesis. They propose optimizing point cloud poses simultaneously through a bundle-adjusting neural lidar field as well as through ICP-inspired geometric optimization. The neural lidar field is supervised with ground truth range images, intensity images, and ray-drop probability images as well as ground truth point clouds. The geometric optimization optimizes poses to minimize a weighted average of the chamfer distances between neighboring lidar frames. The proposed method is validated for lidar novel view synthesis and pose estimation using the Nuscenes and KITTI-360 datasets.

**Strengths:**

1) The authors apply knowledge from nerf pose estimation to the problem of lidar novel view synthesis through the bundle-adjusting lidar nerf.
2) They propose a novel alternating pose optimization between the implicit nerf representation and the explicit point clouds.
3) The proposed method is supported by experiments on two real-world datasets which verify its validity.
4) The paper proposes a new task of jointly recovering pose and lidar point cloud novel views, which is significant for lowering the impact of noisy estimated lidar poses.
5) They achieve state-of-the-art results for lidar novel view synthesis with the proposed method, as well as better point cloud registration.

**Weaknesses:**

The methodology section reads like a work in progress, making it hard to understand the details of the proposed method.
1) There is no intuitive explanation as to why the "J" term in Eq. 5 can simply be dropped. While the main text states that dropping this term allows independent updates of rotation and translation, the supplementary material then states that simultaneous optimization yielded satisfactory results in L527.
2) P is used to denote poses in Eq. 1 as well as point clouds in Eq. 11 and ray-drop loss probabilities in Eq. 13. L173. Similarly, I is used to refer to NeRF images in L112 and then intensity images in Eq. 13. Intensities and ray-drop probabilities are referred to as S and R in L119.
3) L173 states that Eq. 7 shows how CD is in line with each step of ICP, but the optimization steps of ICP are never introduced.
4) Eq. 8 introduces d_clipped which is never used. It takes as parameter a distance "d" that is never introduced.
5) There is no explanation as to how the normals used in Eq. 12 are obtained. Are they obtained from the implicit density field for both predicted and pseudo-groundtruth pointclouds?
6) There is a lot of information in Fig. 3, but the caption does not explain what the figure is trying to show.
7) Sections 3.2 and 3.5 both discuss the nerf optimization, but are separated by sections 3.3 and 3.4 which discuss geometric optimization.

**Questions:**

Please refer to the weakness section.

**Limitations:**

There is no discussion about the optimization time, which might be significantly increased due to the geometric optimization requiring computing the chamfer distance between all combinations of the 4 adjacent frames.

---

> ### Author Rebuttal · Authors · 2024-08-07
>
> We greatly appreciate your positive feedback on our method's novelty and the significance of our new task to reduce the impact of noisy estimated LiDAR poses during LiDAR NVS. The issues you raised have been effectively addressed as follows. Regarding weaknesses 2,5 and your concern about the time-consuming limitation, please kindly refer to the global rebuttal.
>
> **[Q1] The Issue of Term J in Pose Representation Formula.** Commonly used $\xi \in \mathfrak{se}(3)$ represents the transformation in a coupled manner, the translation component of $T$ will be influenced by the rotation parameter $\phi \in \mathfrak{so}(3)$, as shown in Eq. (5) and L153-154. This means the translation of the rigid body's centroid is affected by the rotation.
> Let $B$ and $A$ denote the transformed and original point clouds, where $\overline{B}$ and $\overline{A}$ represent the homogeneous coordinates. Then, $\overline{B} = T\overline{A}$. By omitting $J$ in Eq. (5), according to the principles of matrix multiplication, our method for rigid body pose transformation $\overline{B} = T_{\textbf{w/o} J}\overline{A}$  equates to $B = \boldsymbol{R}A + \rho = \sum_{n=0}^{\infty} \frac{1}{n!}\left(\phi^{\wedge}\right)^n A+\rho$, $\phi \in \mathfrak{so}(3)$ and $\rho \in \mathbb{R}^3$ are the pose parameters to be updated, and the translation is represented solely by $\rho \in \mathbb{R}^3$. **In this way, the rotation parameter $\phi$ and the translation parameter $\rho $ represent the rigid body motion as rotation about the center of mass and translation of the center of mass.** This method represents rotation and translation separately and independently, which is reasonable and physically meaningful.
>
> Besides, in the context of “independently” in L157 of the main text, we refer to the fact that our approach optimizes rotation and translation simultaneously, but their updates are decoupled and do not interfere with each other. Thus, there is no conflict between the Appendix and the main text. We fully understand your concern and believe that changing “independently” to “in a decoupled manner” would be more appropriate.
>
> **[Q3] The Issue of Missing Descriptions in ICP Algorithms.** Thanks for your comments.
> By describing “in line with the optimization objective of each step in the ICP algorithm"(L173), we intend to state that our CD-based optimization is inspired by the well-known classic ICP algorithm. We do not use ICP in our method, and the core optimization step is clearly explained in Eq. (7) and L190-192 (through gradient descent). Given the fluency of description, **we briefly summarize the optimization steps of ICP in L163-L166 and Eq. (6) in our main text.** We will include the detailed optimization steps of ICP in the supplementary material.
>
> **[Q4]: The Issue of Distance Representation ($d$) in Eq. (8).** We apologize for not having explained our method more clearly. $d$ denotes the distance between a pair of matching nearest neighbor points. We intend to use the subscript “clipped" to indicate that this distance has been truncated to be greater than a predefined threshold. Therefore, $d_{clipped}$ represents the $d_i$ we actually used in the first formula. We will update $d_{clipped}$ to $d_i$. The revised formula is as follows:
>
> $w_{i} = \frac{\mathrm{exp}(t/d_i)}{\sum_{i=1}^N \mathrm{exp}(t/d_i)}, t=\mathrm{scheduler}(t_0),d_i=\mathrm{max}(d_0,d)$
>
> where $d$ denotes the distance between a pair of matching nearest neighbor points, $t$ is the temperature to sharpen the distribution of the $d_i$, and $d_0$ is a threshold we set.
>
> **[Q6]: The Issue of Caption Content of Fig. (3).** Based on your suggestion, we will further revise the caption for better readability. The revised caption of Fig. (3) is “**Graph-based RCD (left).** We introduce control factor $t$ in CD to diminish the weighting of non-overlapping regions between point clouds. **Geo-optimizer and its impact on pose optimization (right).** Pose errors are rapidly reduced after each increase caused by NeRF's incorrect optimization direction. Comparison of (a) and (b) shows Geo-optimizer prevents incorrect pose optimization of NeRF.".
>
> The content shown on the left side of Fig. (3) is explained in L185-190. As for the right side of Fig. (3), our primary intention is to provide readers with a clear and intuitive comparison through the figures. To facilitate this, we have carefully annotated the figures with arrows and text for better guidance. Reviewer fgnm and Reviewer TE2m also point out that this figure is informative and accurately demonstrates the effectiveness of our method. We hope the revised title will better explain the information in the figure.
>
> **[Q7]: The Issue of Organizational Structure in the Main Text.** We fully understand your perspective on creating a more cohesive structure. However, within the domain of NeRF / pose-free NeRF, it is common practice to place the NeRF optimization loss (Sec 3.5) at the end to better present the overall pipeline, such as Nope-NeRF[6], NFL[24], LiDARNeRF[52].
>
> Therefore, we adopt the following structure: Sec 3.2 discusses two main aspects of pose-free NeRF, $\textit{i.e.}$, the representation of pose and the encoding method of NeRF. This serves as a foundational premise for the subsequent discussions. Specifically, the representation of pose is essential for pose optimization, as the $T \in \text{SE}(3)$ in original NeRF lacks gradient propagation capabilities ($T + \delta T \notin \text{SE}(3)$). And the encoding method is fundamental across all NeRF works. Then we proceed to discuss the optimization of pose (Sec. 3.3) and NeRF (Sec 3.4-3.5). These three sections (3.3-3.5) constitute our primary contributions. Based on your suggestion, we will integrate Sec 3.5 with the NeRF encoding content from Sec 3.2, thereby encapsulating the entirety of the NeRF optimization process within Sec 3.5, which will help to make the paper more cohesive.

---

### Author Rebuttal · Authors · 2024-08-07

We would like to thank all reviewers for their constructive comments on our work. We identified several comments that are common across more than one reviewer. Thus, we highlight them here.

**[Q1]: The issue of variable reuse in Eq.(13).** Sorry for the confusion. Since we introduce geometry-based losses as a constraint on the original Lidar-NeRF 2D-based loss, we focused primarily on the geometric component. This led to an oversight in the 2D-based loss (Eq.(13)), where we neglected to reintroduce previously defined variables.

Based on your suggestions, throughout the text, we will consistently use $\mathcal{S}$ to represent intensity and $\mathcal{R}$ to represent ray-drop probabilities. Therefore, Eq.(13) can be rewritten as:

$\mathcal{L}_{r}(\mathbf{r})=
  \sum _{\mathbf{r} \in \mathbf{R}} (\lambda_d \begin{Vmatrix} \hat{D}(\mathbf{r}) - D(\mathbf{r}) \end{Vmatrix}_1 +
  \lambda_i \begin{Vmatrix} \mathcal{\hat{S}}(\mathbf{r}) - \mathcal{S}(\mathbf{r}) \end{Vmatrix}_2^2 +
  \lambda_p \begin{Vmatrix} \mathcal{\hat{R}}(\mathbf{r}) - \mathcal{R}(\mathbf{r}) \end{Vmatrix}_2^2)$

Additionally, there is also a minor error of variable reuse in Eq.(1), we will use $K$ to represent the pose in Eq.(1). In this way, our variables are now consistent.

**[Q2]: The issue of calculating the normal in Sec 3.5, L222-L223.** We deeply regret not having provided a clearer explanation. As shown in Eq. (12), $\hat{P}$ represents the synthetic point cloud derived from the implicit density field, while $P$ denotes the ground truth point cloud, as mentioned in lines 220-221. Thus, the normal loss is calculated between the synthetic point cloud $\hat{P}$ and the ground truth point cloud $P$ to ensure more accurate normal vectors of the point cloud synthesized from NeRF. We apologize for the incorrect citation of Eq.(9), the definitions of $\hat{P}$ and $P$ should correspond to those in Eq.(11).

Besides, the calculation of normal vectors involves searching for neighboring points around a point and using Principal Component Analysis to perform plane fitting and determine the normal vector. This process is implemented in many point cloud processing libraries, such as PCL and PyTorch 3D.

**[Q3]: The issue of run-time comparison and analysis.**
1. Run-time comparison. We provide the comparison in the table below.
||| GeoNLF | HASH-LN | BARF-LN | GeoTrans-assisted |
|-|-|-|-|-|-|
|Opt.| sec/epoch |4.06|3.46|10.69|3.45|
|Opt.| Total|~2.0h|~1.7h|~5.0h|~1.2h|
|Infer|sec/frame|0.24|0.24|0.67|0.23|

It is important to note that the other two pose-free baselines (HASH-LN and BARF-LN) completely fail on many sequences, unable to obtain accurate poses and usable reconstruction results. Regarding the GeoTrans-assisted LiDAR-NeRF, despite its short processing time(It uses the same NeRF as HASH-LN, but converges faster because it does not require pose updates), it requires a pre-trained model, which takes days to train and is limited by the training dataset, and still fails on some sequences. Although GeoNLF typically requires a longer time, our method achieves a better balance between accuracy and efficiency. GeoNLF demonstrates greater robustness and has practical value, successfully aligning and reconstructing all sequences.

2. Run-time analysis.

(1) Selective-Reweighting(SR) strategy: Our SR strategy does not introduce any additional time burden. It simply involves selecting the top-k frames based on the training loss from the previous epoch and adjusting their learning rates according to Eq. (11).

(2) Geometric constraints: Normal-based loss and CD are computed between the nearest points of two given point clouds. Since Normal uses the correspondences from CD, we only perform the nearest neighbor search once. Consequently, this approach is time-efficient with 0.026s/frame and 0.8s/epoch. We optimize every 5 epochs, taking only 5 minutes (within 4.2\% of the total time).

(3) Geo-optimizer: We construct a graph where each frame is connected to the two preceding and two following frames. Given that the computation of CD is symmetric, meaning the CD between point cloud A and point cloud B is the same as that between B and A, each graph consisting of 32 frames will have 61 edges for CD computation, with each frame connected to its four neighboring frames. To leverage parallel computation, we downsample each point cloud to the same number of points $N=24000$. We then transform the CD computation into a calculation between two tensors of size $61 \times N \times 3$ based on the graph, allowing us to obtain the entire result in parallel. Consequently, the Geo-optimizer takes only 0.23s/iteration and 16 minutes in total(within 14\% of the total time).

---

> ### Comment · Reviewer_PTin · 2024-08-09
>
> Could you please detail how normals are being computed? I believe this is an important detail to include in the manuscript since the proposed normal loss requires gradients to be backpropagated through the normal estimation function.

---

> > ### Author Response · Authors · 2024-08-10
> > **Computation of normals**
> >
> > We greatly appreciate your suggestions and the insightful comments you provided on the completeness of our paper.
> >
> > Here, we first outline the steps for computing normal vectors:
> > 1. For a point $A$ in the point cloud, we search for its $N$ neighbors (with $N = 50$ in our implementation) and center these points by subtracting the coordinates of $A$, resulting in a matrix $X\in \mathbb{R}^{N\times 3}$.
> > 2. We compute the covariance matrix of this neighborhood ($X^T X$) and perform a differentiable Singular Value Decomposition (SVD) to obtain the eigenvectors and eigenvalues.
> > 3. The normal vector typically corresponds to the eigenvector associated with the smallest eigenvalue of the covariance matrix, representing the local surface orientation.
> >
> > We use the differentiable point cloud library provided by PyTorch3D [1] for normal vector computation, ensuring that the entire process is fully differentiable. We will summarize the normal vector computation process in the main text and provide a detailed discussion in the appendix, with the code to be made publicly available thereafter.
> >
> > [1] Ravi, Nikhila, et al. "Accelerating 3d deep learning with pytorch3d." arXiv preprint arXiv:2007.08501 (2020).

---

### Decision · Program_Chairs · 2024-09-25

**Decision:**

Accept (poster)

**Comment:**

The paper proposes a method for novel view synthesis from Lidar point clouds. It initially received highly discordant ratings (R, WA, A). The rebuttal and subsequent discussion appeared to have addressed the main technical concerns from the more negative reviewer, who decided to lean on a BA. The other two reviewers confirmed their initial ratings.
Based on this, the AC has deemed that a consensus has been found towards acceptance. The AC warmly recommends the authors to incorporate the comments from the reviewers in the camera ready, including those around the clarity/presentation of the method.